# SIRT1-Mediated Epigenetic Protective Mechanisms of Phytosome-Encapsulated *Zea mays* L. var. *ceratina* Tassel Extract in a Rat Model of PM2.5-Induced Cardiovascular Inflammation

**DOI:** 10.3390/ijms26125759

**Published:** 2025-06-16

**Authors:** Wipawee Thukham-Mee, Jintanaporn Wattanathorn, Nut Palachai

**Affiliations:** 1Integrative Complementary Alternative Medicine Research and Development Center in the Research Institute for Human High Performance and Health Promotion, Khon Kaen University, Khon Kaen 40002, Thailand; meewep@gmail.com (W.T.-M.); jinwat05@gmail.com (J.W.); 2Department of Physiology, Faculty of Medicine, Khon Kaen University, Khon Kaen 40002, Thailand; 3Faculty of Medicine, Mahasarakham University, Mahasarakham 44000, Thailand

**Keywords:** cardiovascular diseases, inflammation, oxidative stress, phytosome, tassel, *Zea mays* L. var. *ceratina*, purple waxy corn

## Abstract

Cardiovascular injury caused by fine particulate matter (PM2.5) exposure is an escalating public health concern due to its role in triggering systemic inflammation and oxidative stress. This study elucidates the sirtuin 1 (SIRT1)-mediated epigenetic mechanisms underlying the protective effects of phytosome-encapsulated *Zea mays* L. var. *ceratina* tassel extract (PZT) in a rat model of PM2.5-induced cardiovascular inflammation. Male Wistar rats were pretreated with PZT (100, 200, and 400 mg/kg body weight) for 21 days before and throughout a 27-day PM2.5 exposure period. SIRT1 expression and associated inflammatory and oxidative stress markers were evaluated in cardiac and vascular tissues. The findings revealed that PZT significantly upregulated SIRT1 expression, a key epigenetic regulator known to modulate inflammatory and antioxidant pathways. The activation of SIRT1 inhibited the nuclear factor-kappa B (NF-κB) signaling pathway, leading to a reduction in pro-inflammatory cytokines such as tumor necrosis factor-alpha (TNF-α) and interleukin-6 (IL-6) within cardiac tissue. In vascular tissue, treatment with PZT reduced the levels of tumor necrosis factor-alpha (TNF-α) and transforming growth factor-beta (TGF-β), thereby mitigating inflammatory and fibrotic responses. Furthermore, SIRT1 activation by PZT enhanced the antioxidant defense system by upregulating superoxide dismutase (SOD), catalase (CAT), and glutathione peroxidase (GSH-Px), which was associated with a decrease in malondialdehyde (MDA), a marker of lipid peroxidation. Collectively, these results demonstrate that PZT confers cardiovascular protection through SIRT1-dependent epigenetic modulation, mitigating PM2.5-induced inflammation, oxidative stress, and tissue remodeling. The dual anti-inflammatory and antioxidant actions of PZT via SIRT1 activation highlight its potential as a functional food-based preventative agent for reducing cardiovascular risk in polluted environments.

## 1. Introduction

Cardiovascular diseases (CVDs) are a leading global cause of morbidity and mortality, with growing prevalence linked to lifestyle and environmental pollutants. Fine particulate matter (PM), a common air pollutant from traffic, industry, and biomass burning, is recognized as a significant risk factor for cardiovascular dysfunction, especially in urban and industrial regions [1,2,3].

Due to their small size (<2.5 µm), PM2.5 particles can evade respiratory defenses and enter systemic circulation, inducing oxidative stress and inflammation. These effects contribute to endothelial dysfunction and vascular remodeling, driving conditions such as hypertension and atherosclerosis [4,5]. Although current therapies control symptoms and risk factors, they often do not address these underlying molecular mechanisms, highlighting the need for preventive strategies targeting oxidative and inflammatory pathways activated by PM2.5.

Epigenetic modulation, particularly via SIRT1—a class III histone deacetylase—is crucial in cellular responses to environmental stress. By deacetylating NF-κB, SIRT1 suppresses pro-inflammatory signaling and activates antioxidant defense genes, playing a key role in cardiovascular homeostasis [6,7]. Enhancing SIRT1 activity is thus a promising therapeutic approach against PM2.5-induced cardiovascular damage.

Natural compounds derived from plant sources have shown potential in modulating epigenetic regulators, such as SIRT1. *Zea mays* L. var. *ceratina* (purple waxy corn) is particularly rich in phytochemicals, especially flavonoids, including anthocyanins [8], rutin, quercetin, and kaempferol, which possess strong antioxidant and anti-inflammatory properties. While the kernels of this variety have been extensively studied, the tassel—a traditionally underutilized by-product—also contains a diverse array of bioactive compounds that may contribute to cardiovascular protection [8,9]. However, the clinical potential of these compounds is often limited by poor bioavailability and stability [10]. To overcome these limitations, phytosome encapsulation—a delivery system that complexes bioactive compounds with phospholipids, particularly phosphatidylcholine—has been employed. This approach not only enhances solubility, stability, and intestinal absorption but also facilitates efficient cellular uptake, as phosphatidylcholine is a key component of cell membranes and enables easier passage through lipid bilayers [11,12,13,14].

Building upon prior knowledge, this study investigates the epigenetic protective role of PZT via SIRT1-dependent mechanisms in a rat model of cardiovascular inflammation induced by PM2.5. Male Wistar rats were administered PZT orally for 21 days preceding and throughout a 27-day PM2.5 exposure. Cardiovascular outcomes were assessed by measuring inflammatory markers such as NF-κB, TNF-α, IL-6, and TGF-β, along with oxidative stress indicators including MDA, and the activities of key antioxidant enzymes (SOD, CAT, and GSH-Px). Special emphasis was placed on the role of SIRT1 expression and activation as a central epigenetic regulator modulating these pathological responses.

By uncovering the SIRT1-dependent pathways through which PZT mitigates PM2.5-induced inflammation and oxidative damage, this study highlights the potential of PZT as a novel therapeutic or functional food ingredient for cardiovascular protection. These findings further underscore the value of targeting epigenetic mechanisms in the development of preventive strategies against pollution-related cardiovascular diseases.

## 2. Results

### 2.1. Bioactive Compounds of Zea mays L. var. ceratina Tassel Extract and PZT

The HPLC profiles of *Zea mays* L. var. *ceratina* tassel extract and its phytosome formulation (PZT) are presented in Figure 1. The quantitative analysis of key bioactive compounds is summarized in Table 1.

In the crude tassel extract, aside from the previously reported anthocyanins—particularly cyanidin-3-glucoside [8]—the major detected flavonoids were rutin (0.730 ± 0.012 µg/mg), quercetin (0.077 ± 0.003 µg/mg), and kaempferol (0.042 ± 0.000 µg/mg).

In the PZT formulation, which contains a 1:1 ratio of crude extract to phosphatidylcholine, the concentrations of the same compounds were as follows: rutin (0.602 ± 0.022 µg/mg), quercetin (0.078 ± 0.006 µg/mg), and kaempferol (0.074 ± 0.000 µg/mg). Notably, rutin content showed a slight reduction, quercetin remained nearly unchanged, and kaempferol content was markedly increased compared to the crude extract.

These results suggest that the phytosome formulation effectively retains the principal bioactive compounds of the original extract and may enhance the incorporation or stability of specific constituents, particularly kaempferol. The increase in kaempferol concentration in PZT highlights the potential of the phytosome delivery system to improve the stability or solubility of certain flavonoids. Overall, the extraction and encapsulation techniques employed appear suitable for preserving and potentially enhancing the bioactive profile of *Zea mays* L. var. *ceratina* tassel.

### 2.2. Effect of PZT on SIRT1-Mediated Epigenetic Modifications in Cardiac and Vascular Tissues

Epigenetic modifications, particularly histone modifications and deacetylation, play a pivotal role in the development and progression of cardiovascular diseases by regulating gene expression in response to environmental factors such as oxidative stress and inflammation [15]. To investigate the impact of PZT on these epigenetic changes, we focused on sirtuins, a family of NAD+-dependent class III histone deacetylases, with a particular emphasis on SIRT1. SIRT1 is crucial in cellular responses to oxidative stress and inflammation, both of which are central to the pathophysiology of cardiovascular conditions.

As shown in Figure 2 and Figure 3, exposure to PM2.5 led to a significant reduction in SIRT1 expression in both cardiac and vascular tissues (*p* < 0.05 for cardiac tissue and *p* < 0.001 for vascular tissue, compared to the naïve control group). This decrease suggests that PM2.5 exposure induces epigenetic changes that may impair cellular stress responses, potentially contributing to cardiovascular dysfunction.

In the positive control group, rats administered prednisolone (1 mg/kg BW) exhibited a significant increase in SIRT1 expression in both tissue types. This suggests that prednisolone may counteract the epigenetic effects of PM2.5 exposure, thereby alleviating the systemic inflammation and oxidative stress induced by particulate matter (*p* < 0.05 for cardiac tissue and *p* < 0.001 for vascular tissue, compared to the PM2.5 + vehicle group).

Treatment with PZT at doses of 100, 200, and 400 mg/kg BW also significantly reversed the PM2.5-induced reduction in SIRT1 expression, suggesting a protective effect. Specifically, in cardiac tissue, all doses of PZT resulted in a significant increase in SIRT1 expression (*p* < 0.05, compared to the PM2.5 + vehicle group) (Figure 2). Similarly, in vascular tissue, PZT treatment significantly restored SIRT1 expression at the 200 and 400 mg/kg BW doses (*p* < 0.01 and *p* < 0.001, respectively) (Figure 3). Although the 100 mg/kg BW dose did not show a statistically significant effect, a trend towards improvement was observed.

These findings indicate that PZT exerts a dose-dependent effect in reversing the downregulation of SIRT1 induced by PM2.5 exposure, particularly in vascular tissue. By modulating the expression of key epigenetic regulators like sirtuins, PZT may mitigate the harmful effects of systemic inflammation and oxidative stress, contributing to the preservation of cardiovascular health. The results suggest that the beneficial effects of PZT are likely linked to the modulation of epigenetic factors such as SIRT1, which are crucial for cellular defense mechanisms against environmental stressors.

### 2.3. Effect of PZT on Inflammatory Markers in Cardiac and Vascular Tissues

Environmental pollutants, particularly particulate matter like PM2.5, play a significant role in the pathophysiology of cardiovascular diseases, primarily through the induction of inflammation [16]. PM2.5 particles penetrate deeply into the lungs and enter systemic circulation, exacerbating inflammation in various tissues, including the cardiovascular system. Chronic inflammation, mediated by pro-inflammatory cytokines and transcription factors, is a central contributor to cardiovascular disease development and progression. In this study, we examined the effect of PZT on several key inflammatory markers involved in both cardiac and vascular tissues, aiming to assess its potential therapeutic value in mitigating PM2.5-induced cardiovascular inflammation.

NF-κB functions as a key transcription factor that controls the activation of genes related to inflammation, immunity, and cell survival. Typically, it remains inactive in the cytoplasm by binding to its inhibitor, IκB. Upon stimulation by pro-inflammatory factors like PM2.5, IκB undergoes degradation, allowing NF-κB to move into the nucleus. There, it promotes the transcription of various pro-inflammatory mediators, including cytokines and adhesion molecules.

Figure 4 and Figure 5 show that exposure to PM2.5 resulted in a significant upregulation of NF-κB expression in both cardiac and vascular tissues (*p* < 0.001 compared to naïve controls), indicating the activation of this crucial inflammatory pathway. The vehicle-treated rats exposed to PM2.5 further support that this upregulation is specifically due to PM2.5 exposure, rather than the vehicle itself.

Treatment with prednisolone, a known anti-inflammatory corticosteroid, served as a positive control and significantly reduced NF-κB expression in both cardiac and vascular tissues (*p* < 0.001 compared to the PM2.5 + vehicle group). Prednisolone suppresses NF-κB activation through its glucocorticoid receptor, interacting with the NF-κB pathway at multiple points, thereby reducing the expression of inflammatory genes.

PZT treatment at all tested doses (100, 200, and 400 mg/kg BW) also reduced NF-κB expression in both cardiac and vascular tissues in a dose-dependent manner (*p* < 0.001, compared to the PM2.5 + vehicle group). This suggests that PZT effectively inhibits NF-κB signaling, attenuating the inflammatory cascade induced by PM2.5 exposure.

TNF-α is a highly influential pro-inflammatory cytokine that triggers and intensifies inflammatory processes. It stimulates multiple signaling cascades that contribute to inflammation, impair endothelial function, and cause tissue injury. Within the cardiovascular system, increased TNF-α concentrations are linked to vascular inflammation, the development of atherosclerosis, and the progression of heart failure.

Figure 4 and Figure 5 illustrate that vehicle-treated rats exposed to PM2.5 exhibited significantly elevated TNF-α levels in both cardiac and vascular tissues (*p* < 0.001 compared to naïve controls), highlighting the inflammatory response elicited by particulate matter. Prednisolone treatment significantly reduced TNF-α levels in both cardiac and vascular tissues (*p* < 0.001 compared to the PM2.5 + vehicle group), consistent with the well-known anti-inflammatory properties of corticosteroids.

Similarly, PZT treatment resulted in a significant reduction in TNF-α expression in both cardiac and vascular tissues. In cardiac tissue, TNF-α levels were markedly decreased at all doses (*p* < 0.05, 0.01, and 0.001, respectively, compared to the PM2.5 + vehicle group). In vascular tissue, PZT treatment significantly reduced TNF-α expression at 200 and 400 mg/kg BW (*p* < 0.01 and *p* < 0.001, respectively, compared to the PM2.5 + vehicle group). Although the 100 mg/kg BW dose did not significantly reduce TNF-α levels, a trend toward decreased levels was observed. These results suggest that PZT exerts its anti-inflammatory effects, at least in part, by modulating TNF-α levels, reducing cardiac and vascular inflammation induced by PM2.5.

IL-6 is a vital cytokine that contributes significantly to inflammation by stimulating the production of acute-phase proteins, including C-reactive protein (CRP). Increased IL-6 concentrations are linked to chronic inflammatory diseases, especially those involving the cardiovascular system. When exposed to PM2.5, IL-6 plays a central role in triggering the body’s systemic inflammatory reaction.

Figure 4 shows that IL-6 levels were significantly elevated in the cardiac tissue of PM2.5-exposed rats treated with the vehicle (*p* < 0.001 compared to naïve controls). Treatment with both prednisolone and all doses of PZT (100, 200, and 400 mg/kg BW) significantly reduced IL-6 expression (*p* < 0.001, 0.01, 0.01, and 0.001, respectively, compared to the PM2.5 + vehicle group). This reduction indicates that PZT can modulate the systemic inflammatory response, preventing excessive IL-6 production, which contributes to cardiovascular damage.

TGF-β is a versatile cytokine that regulates cell growth, differentiation, and tissue regeneration. It also serves as a major factor in the fibrosis and remodeling of blood vessels, processes that underlie the lasting effects of chronic inflammation. Within vascular tissues, TGF-β is instrumental in driving the fibrosis and thickening of the intimal layer, key features observed in atherosclerosis and various vascular disorders.

Figure 5 illustrates that PM2.5 exposure significantly increased TGF-β expression in the vascular tissue of vehicle-treated rats (*p* < 0.001 compared to naïve controls). This elevation suggests that PM2.5 exposure promotes not only acute inflammatory responses but also long-term vascular remodeling. Treatment with prednisolone and all doses of PZT (100, 200, and 400 mg/kg BW) significantly reduced TGF-β levels (*p* < 0.001, 0.05, 0.01, and 0.001, respectively, compared to the PM2.5 + vehicle group). This reduction suggests that PZT may help prevent or mitigate vascular remodeling and fibrosis, critical processes in the progression of cardiovascular diseases.

In summary, treatment with PZT significantly attenuated PM2.5-induced inflammation by reducing the expression of key inflammatory markers, including NF-κB, TNF-α, and IL-6 in cardiac tissues, as well as NF-κB, TNF-α, and TGF-β in vascular tissues. The most pronounced anti-inflammatory effects were observed at higher PZT doses. Prednisolone also reduced these markers, supporting the anti-inflammatory potential of PZT. These findings highlight the efficacy of PZT in mitigating PM2.5-induced cardiovascular inflammation.

### 2.4. Effect PZT on Oxidative Stress in Cardiac and Vascular Tissues

Given the critical role of oxidative stress in the development of cardiovascular diseases associated with PM2.5 exposure [17], we evaluated the impact of PZT treatment on oxidative stress markers in both cardiac and vascular tissues.

As shown in Figure 6, exposure to PM2.5 significantly reduced the activities of antioxidant enzymes—SOD, CAT, and GSH-Px—in cardiac tissue (all *p* < 0.001 compared to naïve controls). In contrast, prednisolone treatment significantly restored the activity of these enzymes (all *p* < 0.05 compared to the PM2.5 + vehicle group). PZT treatment, particularly at doses of 200 and 400 mg/kg BW, further enhanced the activities of all of these enzymes (all *p* < 0.001 compared to the PM2.5 + vehicle group). Notably, PZT at 100 mg/kg BW showed a significant increase in only CAT activity (*p* < 0.05 compared to the PM2.5 + vehicle group), suggesting a dose-dependent improvement in antioxidant defense mechanisms within cardiac tissue.

In Figure 7, PM2.5 exposure also significantly decreased the activities of CAT, SOD, and GSH-Px in the vascular tissue of vehicle-treated rats (all *p* < 0.001 compared to naïve controls). Prednisolone treatment notably reversed the reductions in CAT and SOD activities (both *p* < 0.05 compared to the PM2.5 + vehicle group). Similarly, PZT at all tested doses (100, 200, and 400 mg/kg BW) significantly increased the activities of CAT (*p* < 0.05, *p* < 0.001, and *p* < 0.001, respectively), SOD (*p* < 0.05, *p* < 0.001, and *p* < 0.001, respectively), and GSH-Px (*p* < 0.01, *p* < 0.001, and *p* < 0.001, respectively), demonstrating a strong dose-dependent response in vascular tissue.

To further confirm the antioxidant-enhancing effects of PZT, we assessed MDA levels, an indicator of lipid peroxidation. As depicted in Figure 8, PM2.5 exposure significantly elevated MDA levels in both cardiac and vascular tissues of vehicle-treated rats (*p* < 0.001 compared to naïve controls). Treatment with prednisolone and PZT at doses of 200 and 400 mg/kg BW significantly reduced MDA levels in cardiac tissue (*p* < 0.05, *p* < 0.001, and *p* < 0.001, respectively, compared to the PM2.5 + vehicle group). Interestingly, PZT at 100 mg/kg BW also reduced MDA levels in cardiac tissue, though the effect was less pronounced than at higher doses. In vascular tissue, both prednisolone and all doses of PZT significantly lowered MDA levels (*p* < 0.001 for all comparisons to the PM2.5 + vehicle group).

In conclusion, PM2.5 exposure led to significant increases in MDA levels, reflecting enhanced lipid peroxidation, and reduced activities of antioxidant enzymes in both cardiac and vascular tissues. Treatment with prednisolone and PZT effectively mitigated oxidative stress by reducing MDA levels and enhancing the activity of antioxidant enzymes, with the most significant effects observed at higher PZT doses. The dose-dependent nature of PZT’s effects highlights its potential in restoring the balance between pro-oxidant and antioxidant factors in cardiovascular tissues, offering protection against oxidative damage induced by PM2.5 exposure. These findings emphasize the therapeutic potential of PZT in combating oxidative stress and mitigating PM2.5-induced cardiovascular injury.

## 3. Discussion

This study presents comprehensive insights into the protective effects of PZT in a PM2.5-induced rat model of systemic inflammation. PM2.5 exposure, a known environmental risk factor, accelerates cardiovascular disease through complex and interlinked mechanisms, including oxidative stress, inflammation, and epigenetic alterations. Each of these mechanisms plays a distinct yet interconnected role in the progression of pollutant-induced cardiovascular damage [18,19,20]. Our findings reveal that PZT mitigates these harmful effects by modulating epigenetic pathways, suppressing inflammation, and enhancing antioxidant defenses, suggesting a multi-faceted therapeutic potential for cardiovascular protection in polluted environments.

Epigenetic mechanisms, particularly the regulation of sirtuins, play a critical role in helping cells adapt to environmental stressors and form a key part of the body’s defense against oxidative and inflammatory damage [21,22]. Sirtuins, such as SIRT1, are NAD+-dependent deacetylases that influence transcription, modulate cellular aging, and respond to stress by altering gene expression [23]. Within cardiovascular health, SIRT1 is essential for preserving endothelial function, guiding vascular remodeling, and maintaining metabolic balance. By activating endothelial nitric oxide synthase (eNOS), SIRT1 boosts nitric oxide (NO) production, which supports vasodilation, regulates blood flow, and reduces the risk of hypertension [24]. It also combats endothelial inflammation by suppressing NF-κB signaling and lowering adhesion molecule expression, thereby protecting against vascular inflammation and atherosclerosis [25]. In vascular remodeling, SIRT1 prevents excessive proliferation and the migration of smooth muscle cells, enabling adaptive structural changes without pathological narrowing or stiffening [26]. Metabolically, SIRT1 governs lipid and glucose homeostasis via PGC-1α and liver X receptors (LXRs), enhancing mitochondrial function, promoting cholesterol efflux, and reducing foam cell formation [27]. Additionally, SIRT1 mitigates oxidative stress by increasing antioxidant enzyme activity, shielding cells from age-related damage and apoptosis [28]. Disruptions in SIRT1 function, such as those triggered by PM2.5 exposure, weaken these protective mechanisms, accelerating cardiovascular dysfunction and disease progression [29,30,31]. PM2.5 exposure, as shown in this study, significantly downregulated SIRT1 levels in cardiac and vascular tissues. This decline is likely due to the overwhelming oxidative and inflammatory burden, which exhausts cellular adaptive capacity, resulting in increased tissue damage and reduced resilience.

The administration of PZT at doses of 100, 200, and 400 mg/kg BW effectively restored SIRT1 expression to levels comparable to those observed in naïve controls, with the highest dose showing the most pronounced recovery. This suggests that PZT may exert its effects through epigenetic modulation, enhancing cellular resilience in response to pollutant-induced stress. By stabilizing SIRT1 expression, PZT potentially prevents further downstream epigenetic changes that contribute to inflammation and oxidative stress, thus interrupting the positive feedback loop that perpetuates cardiovascular injury. The ability of prednisolone to also increase SIRT1 levels underscores the clinical significance of targeting this pathway for therapeutic intervention in environmental cardiovascular health.

Inflammation plays a crucial role in the pathological development of cardiovascular diseases exacerbated by PM2.5 exposure. NF-κB, a key transcription factor, is rapidly activated by environmental stressors like PM2.5 and regulates the expression of pro-inflammatory cytokines such as TNF-α, IL-6, and TGF-β [32]. These cytokines are central to chronic inflammation, tissue remodeling, and fibrosis, all of which contribute to cardiovascular dysfunction. Our study demonstrated that PM2.5 exposure resulted in a significant activation of NF-κB in both cardiac and vascular tissues, leading to the upregulation of TNF-α, IL-6, and TGF-β. This upregulation triggers a cascade of inflammatory responses that drive both acute and chronic cardiovascular damage [33,34].

PZT treatment proved highly effective in attenuating NF-κB activation and cytokine levels in both cardiac and vascular tissues. Notably, PZT significantly reduced TNF-α and IL-6 levels in cardiac tissue, and TNF-α and TGF-β levels in vascular tissue. This reduction in inflammatory markers suggests that PZT may alleviate the chronic inflammation induced by PM2.5, potentially reducing long-term tissue remodeling and fibrosis. Importantly, the anti-inflammatory effects of PZT were dose-dependent, with higher doses (200 and 400 mg/kg BW) showing more significant reductions in these inflammatory markers, particularly in vascular tissue. The dose-dependent nature of these effects highlights the importance of optimal dosing to achieve maximum therapeutic benefits in mitigating inflammation caused by environmental pollutants.

The broad-spectrum anti-inflammatory action of PZT, targeting multiple inflammatory mediators, is crucial for its ability to stabilize vascular function during prolonged pollutant exposure. The similar effects of prednisolone, a well-known corticosteroid, further support the therapeutic potential of PZT in reducing cardiovascular inflammation. By inhibiting NF-κB activation through its interaction with the glucocorticoid receptor, prednisolone reduces the expression of inflammatory cytokines [35]. The fact that PZT exhibited comparable effects strengthens its potential as a promising therapeutic agent. These findings suggest that PZT’s effects may extend beyond immediate cytokine suppression, potentially altering inflammatory gene expression and providing long-term resilience against pollutant-induced stress.

Cardiovascular injury triggered by PM2.5 exposure is closely linked to oxidative stress, which occurs when reactive oxygen species (ROS) are produced in excess and overwhelm the antioxidant defense systems. These ROS—such as superoxide anions and hydrogen peroxide—are highly reactive and can compromise cellular components, including lipids, proteins, and nucleic acids, leading to structural and functional damage [36,37]. This oxidative damage is a precursor to cellular dysfunction, activating apoptotic pathways and triggering cellular senescence. The elevated ROS levels caused by PM2.5 exposure compromise cellular integrity and accelerate the onset of cardiovascular pathology. Our study demonstrated that PM2.5 exposure significantly increased MDA levels, a widely recognized marker of lipid peroxidation, while reducing the activity of essential antioxidant enzymes, including SOD, CAT, and GSH-Px. This imbalance in oxidative and antioxidative forces exacerbates tissue damage and accelerates structural and functional cardiovascular deterioration [37,38,39].

PZT treatment showed a promising ability to restore antioxidant enzyme activities, particularly at the 200 and 400 mg/kg BW doses, and reduced MDA levels significantly. These results indicate that PZT may play a crucial role in re-establishing redox homeostasis in cardiovascular tissues exposed to PM2.5. The dual strategy of PZT, which likely involves both the direct scavenging of ROS and an enhancement of the body’s endogenous antioxidant defenses, provides robust protection against oxidative damage. By reducing lipid peroxidation and protecting cellular macromolecules from ROS-induced damage, PZT helps preserve cardiovascular tissue integrity. This suggests that PZT not only prevents the immediate effects of oxidative damage but may also promote long-term resilience against ongoing pollutant-induced oxidative stress.

The dose-dependent nature of PZT’s antioxidant effects further emphasizes the significance of appropriate dosing to achieve maximal protective benefits. While the 200 and 400 mg/kg BW doses were effective in restoring antioxidant enzyme activities and reducing oxidative stress, the 100 mg/kg BW dose was insufficient to fully restore SOD and GSH-Px activities. This observation suggests that the antioxidant effects of PZT may be dose-dependent, with higher doses providing more substantial benefits in combating oxidative damage. It is crucial to identify the optimal dose to fully harness the protective effects of PZT and ensure its clinical relevance as a therapeutic strategy.

Furthermore, the similar enhancement of antioxidant enzyme levels by prednisolone reinforces the notion that bolstering antioxidant defenses is a viable approach for mitigating oxidative stress and subsequent cardiovascular injury. This suggests that PZT may have therapeutic potential comparable to well-established treatments, like prednisolone, in counteracting the adverse effects of environmental pollutants such as PM2.5. Together, these findings highlight the importance of antioxidant interventions in addressing the oxidative burden placed on cardiovascular tissues by PM2.5 and other environmental stressors, with PZT emerging as a promising candidate for such interventions.

The interplay between PZT’s effects on oxidative stress, inflammation, and epigenetic modulation establishes a synergistic mechanism in which these pathways reinforce one another, ultimately preserving cardiovascular function. A key component of this synergy is the restoration of SIRT1 expression, which plays a crucial role in mitigating both oxidative and inflammatory stress. SIRT1, a NAD+-dependent deacetylase, regulates various cellular processes, including gene expression, through epigenetic modifications. It exerts protective effects by modulating the acetylation of transcription factors such as NF-κB, a central mediator of the inflammatory response.

In the context of PZT treatment, the restoration of SIRT1 expression enhances cellular resistance to oxidative stress. This, in turn, reduces NF-κB activation, which is often upregulated in response to environmental stressors like PM2.5. By limiting the activation of NF-κB, PZT curbs the production of pro-inflammatory cytokines such as TNF-α, IL-6, and TGF-β, which are implicated in chronic inflammation and tissue remodeling. These effects create a feedback loop, where reduced oxidative stress and inflammation work together to suppress further cytokine production, mitigating both acute and chronic cardiovascular damage.

Moreover, enhanced antioxidant defenses also play a crucial role in this mechanism. By reducing levels of ROS, PZT not only protects cellular structures but also prevents the ROS-mediated activation of inflammatory pathways. ROS are known to be potent activators of NF-κB and other pro-inflammatory transcription factors, contributing to a vicious cycle of inflammation and tissue damage. By reducing ROS levels, PZT breaks this cycle, offering protection against both immediate and cumulative effects of PM2.5 exposure.

Together, these mechanisms—epigenetic modulation via SIRT1, the reduction of oxidative stress, and the inhibition of inflammatory pathways—provide a comprehensive defense against pollutant-induced cardiovascular injury. This multi-faceted approach suggests that PZT may not only slow the progression of cardiovascular diseases exacerbated by environmental pollutants like PM2.5 but could also potentially reverse some of the damage, offering a promising therapeutic avenue for individuals exposed to prolonged environmental stressors.

A crucial aspect of PZT’s therapeutic potential lies in its bioactive composition, as characterized through HPLC analysis. The formulation retained key flavonoids from the *Zea mays* L. var. *ceratina* tassel extract, including rutin, quercetin, and kaempferol. While the phytosome formulation exhibited a slight reduction in rutin concentration, quercetin remained stable, and kaempferol was present at a higher concentration compared to the crude extract. This suggests that the phytosome complex not only preserves most bioactive compounds but may also enhance the incorporation or stability of certain constituents, such as kaempferol, which could contribute to its pharmacological effects.

Flavonoids such as rutin and quercetin possess well-documented antioxidant and anti-inflammatory properties, which likely contribute to PZT’s ability to mitigate PM2.5-induced cardiovascular damage [40,41]. Rutin has been shown to scavenge ROS and upregulate endogenous antioxidant enzymes, aligning with PZT’s observed ability to restore SOD, CAT, and GSH-Px activity. Similarly, quercetin exerts protective effects by inhibiting NF-κB activation, thereby reducing the expression of pro-inflammatory cytokines such as TNF-α and IL-6, both of which were significantly downregulated following PZT administration. The presence of kaempferol further strengthens PZT’s therapeutic profile, as this flavonol has been reported to regulate endothelial function by modulating nitric oxide bioavailability and suppressing vascular inflammation [42].

Importantly, quercetin, rutin, and kaempferol have been linked to epigenetic modulation, particularly through the regulation of SIRT1 expression and activity. Quercetin is a well-documented SIRT1 activator, enhancing its deacetylase function, which plays a crucial role in cellular stress resistance and anti-inflammatory responses [43,44]. Similarly, rutin has been shown to activate SIRT1, promoting mitochondrial biogenesis and metabolic homeostasis [45]. Kaempferol also upregulates SIRT1 expression, contributing to oxidative stress regulation, mitochondrial function, and endothelial protection [46,47]. Given that SIRT1 is essential in counteracting PM2.5-induced epigenetic dysregulation, the presence of these flavonoids in PZT suggests a potential mechanism by which the formulation exerts its protective effects.

The phytosome encapsulation process likely enhances the solubility and cellular uptake of these bioactive compounds, thereby improving their bioavailability and therapeutic efficacy. This is supported by the observation that PZT exhibited a more pronounced protective effect compared to the crude extract in preliminary in vitro studies [8]. Collectively, these findings suggest that the phytosome-based delivery system effectively preserves and optimizes the bioactive profile of *Zea mays* L. var. *ceratina* tassel extract, providing a strong mechanistic basis for its observed anti-inflammatory, antioxidant, and epigenetic regulatory effects.

In summary, our results demonstrate that PZT mitigates PM2.5-induced cardiovascular injury via a multi-targeted mechanism involving oxidative stress reduction, inflammation suppression, and epigenetic modulation through SIRT1 activation. However, several limitations should be acknowledged. First, this study used only male rats to minimize biological variability, future studies should include both sexes to investigate sex-specific responses. Second, while the 21-day PM2.5 exposure regimen simulates subacute real-world exposure and aligns with previous models, longer durations are needed to better capture chronic pathological effects. Third, although we focused on SIRT1 due to its well-established relevance to oxidative and inflammatory pathways and its interaction with dietary flavonoids, future studies should explore other sirtuins and epigenetic regulators. Fourth, the study did not include pharmacokinetic or systemic bioavailability assessments of the key bioactive compounds, which are critical for confirming enhanced delivery and informing dose translation. Lastly, the lack of direct cardiovascular functional measurements (e.g., blood pressure or echocardiography) limits the translation of molecular findings into clinical relevance; future studies should incorporate such evaluations.

Taken together, our findings provide a strong mechanistic basis for the protective role of PZT in PM2.5-induced injury and offer promising implications for its use as a preventive or therapeutic strategy in populations exposed to environmental air pollutants. Future investigations should prioritize pharmacokinetic profiling, sex-specific analyses, functional cardiovascular endpoints, and extended exposure durations to support translational and clinical development.

## 4. Materials and Methods

### 4.1. Preparation of PZT

The tassels of *Zea mays* L. var. *ceratina* (voucher specimen KKU No. 25979) were authenticated and analyzed for heavy metals and hazardous substances to ensure safety. Following verification, the tassels were thoroughly washed and dried under controlled conditions at 60 °C for 72 h using a Memmert oven (Memmert GmbH + Co. KG, Schwabach, Germany). The dried tassels were then finely ground into powder. 

A 50% hydroalcoholic solution was used to extract the powdered tassel material through maceration. Following this process, the mixture was centrifuged at 3000 rpm for 10 min to separate solid residues. The resulting supernatant was filtered using Whatman No. 1 paper, concentrated with a rotary evaporator, and then freeze-dried into a fine powder using a Labconco freeze dryer (Labconco Corporation, Kansas City, MO, USA).

Phytosome preparation was carried out using a modified method adapted from earlier work [8,13]. Phosphatidylcholine was first dissolved in dichloromethane, while the hydroalcoholic plant extract was diluted in 50% ethanol. The two solutions were then mixed and exposed to ultrasonic treatment for three cycles (each lasting two minutes) at a frequency of 25–30 kHz to initiate vesicle formation. The blend was subsequently stirred at 25 °C for eight hours to allow for the organized assembly of lipid bilayers.

Residual solvents were removed through rotary evaporation at 45 °C for three hours. The final phytosome preparation was then spray-dried using a BUCHI Mini Spray Dryer (B-290, BÜCHI Labortechnik AG, Switzerland). To ensure stability and prevent moisture absorption, the spray-dried product was stored at 4 °C in a desiccator containing silica gel.

### 4.2. Chromatographic Fingerprint

To establish the fingerprint chromatogram, a high-performance liquid chromatography (HPLC) analysis was conducted using a Waters^®^ system paired with a 2998 photodiode array detector. Separation was carried out on a Purospher^®^ STAR C-18 encapped column (5 µm, Li-ChroCART^®^ 250-4.6) equipped with a Sorbet HPLC cartridge (Lot No. HX255346; Merck KGaA, Darmstadt, Germany). The mobile phase consisted of acetonitrile (solvent A) and 0.1% formic acid in deionized water (solvent B), both sourced from Thermo Fisher Scientific (Waltham, MA, USA). Elution was performed at 1.0 mL/min following a gradient program: 75% B (0–10 min), 40% B (10–20 min), 30% B (20–30 min), and 20% B (30–35 min). Prior to injection, samples were filtered through a 0.45 µm Millipore membrane (MilliporeSigma, Burlington, MA, USA), and 20 µL was injected into the system. Detection was carried out at multiple wavelengths (254, 275, 310, 320, and 370 nm), and data were processed using Empower™ 3 software, Feature Release 4 Hotfix 1 (Waters Corporation, Milford, MA, USA).

### 4.3. Experimental Protocols

To investigate the protective properties of PZT, an in vivo model simulating PM2.5-induced oxidative stress and systemic inflammation was employed. Eight-week-old male Wistar rats (180–220 g) were sourced from Nomura Siam International Co., Ltd., Bangkok, Thailand, and maintained in standard laboratory conditions—six per metal cage—with controlled temperature (23 ± 2 °C), a 12 h light/dark cycle, and unrestricted access to food and water. All experimental procedures were conducted in accordance with ethical guidelines and were approved by the Institutional Animal Ethics Committee of Khon Kaen University (Approval No. IACUC-KKU 11/64).

After a one-week acclimatization period, the rats were randomly assigned to six groups (n = 6):Group I (naïve control): This group received only the vehicle (0.5% carboxymethylcellulose in distilled water) and a standard diet.Group II (PM2.5 + vehicle): This group received the vehicle orally for 21 days prior to and throughout the 27-day PM2.5 exposure.Group III (PM2.5 + prednisolone): This group received prednisolone (1 mg/kg BW) suspended in the vehicle, administered orally for 21 days before and during PM2.5 exposure, serving as the positive control.Groups IV–VI (PM2.5 + PZT100, PZT200, and PZT400): This group received oral doses of PZT at 100, 200, and 400 mg/kg BW, respectively, suspended in the vehicle and administered orally for 21 days prior to and throughout the PM2.5 exposure period.

Treatments were given once daily throughout both the 21-day pre-exposure period and the subsequent 27-day PM2.5 exposure phase. Body weight, food intake, and water consumption were monitored daily over the course of the study. At the conclusion of the experiment, cardiac and vascular tissues were collected for biochemical analyses. Cardiac tissue was examined for SIRT1 expression and pro-inflammatory markers, including NF-κB, TNF-α, and IL-6. In vascular tissues, levels of NF-κB, TNF-α, and TGF-β were analyzed. Furthermore, antioxidant enzyme activities—SOD, CAT, and GSH-Px—as well as MDA levels, an index of lipid peroxidation, were measured. An overview of the experimental design is illustrated in Figure 9.

### 4.4. PM2.5-Induced Systemic Inflammation Model

The systemic inflammation and oxidative stress model induced by PM2.5 exposure was developed by modifying the methodology outlined by Sioutas et al. [8,48]. Male Wistar rats were housed in a transparent plastic chamber (30.5 × 52.5 × 17 cm) equipped with an air filtration system to maintain controlled environmental conditions. A PM2.5 suspension (Sigma-Aldrich, St. Louis, MO, USA; product no. NIST1648A) was aerosolized using a pressure-cycled ventilator (World Precision Instruments, Sarasota, FL, USA; model CW-SAR-830/AP).

To ensure accurate and consistent exposure, a particle counter (model HT-9600/HTI, Tübingen, Germany) continuously monitored the PM2.5 concentration, maintaining it at or above 100 µg/m^3^—approximately ten times the ambient PM2.5 concentration. For 21 days, rats were exposed to the PM2.5 aerosol for 3 h each day, five days per week. This exposure regimen mimics environmental and occupational conditions, simulating chronic, subacute exposure typical of real-world scenarios.

A two-day break each week allowed the rats a recovery period, reflecting realistic exposure patterns, while the 3 h daily exposure ensured a sufficient dose to induce systemic effects without causing acute toxicity. The 21-day duration was critical for assessing the cumulative impact of PM2.5 on systemic inflammation and oxidative stress.

### 4.5. Assessment of Oxidative Stress in Cardiac and Vascular Tissues

To evaluate oxidative stress in cardiac and vascular tissues, assays targeting lipid peroxidation and antioxidant enzyme activities were conducted. Tissue homogenization was performed using 10 mg of sample per 50 µL of 0.1 M potassium phosphate buffer (pH 7.4). Protein concentrations in the homogenates were determined by measuring absorbance at 280 nm using a NanoDrop 2000c spectrophotometer (Thermo Fisher Scientific) [49].

#### 4.5.1. SOD Activity

SOD activity was evaluated based on its capacity to catalyze the conversion of superoxide anions (O_2_^−^) into molecular oxygen (O_2_) and hydrogen peroxide (H_2_O_2_). The assay utilized the following reagents from Sigma-Aldrich: 0.2 M phosphate buffer (pH 7.8), 0.01 M EDTA, 15 µM cytochrome C, and 0.5 mM xanthine (pH 7.4). A total of 200 µL of the reaction mixture was prepared, followed by the addition of 20 µL xanthine oxidase (0.90 mU/mL) and 20 µL of the tissue homogenate. The reaction’s progress was monitored by measuring absorbance at 415 nm. SOD activity was quantified against a standard curve (0–25 U/mL) and reported as units per milligram of protein [50].

#### 4.5.2. CAT Activity

CAT activity was determined based on its ability to decompose H_2_O_2_. The assay involved mixing 50 µL of 30 mM H_2_O_2_ in 50 mM phosphate buffer (pH 7.0) with 10 µL of the tissue homogenate. The reaction was halted by adding 150 µL of 5 mM potassium permanganate followed by 25 µL of 4 M sulfuric acid. Absorbance was read at 490 nm, and CAT activity was calculated using a standard curve (10–100 U/mL), with results expressed as units per milligram of protein [51].

#### 4.5.3. GSH-Px Activity

GSH-Px activity was assessed by incubating 20 µL of tissue homogenate with a reaction mixture containing 10 µL of 1 mM dithiothreitol (DTT), 10 mM monosodium phosphate, 1 mM sodium azide, 100 µL of 40 mM phosphate buffer (pH 7.0), 50 mM reduced glutathione (GSH), and 30% hydrogen peroxide. The mixture was incubated at 25 °C for 10 min, after which 10 µL of 10 mM DTNB was added. Absorbance was measured at 412 nm. GSH-Px activity was calculated using a standard curve (0–5 U/mL) and reported as units per milligram of protein [52].

#### 4.5.4. Lipid Peroxidation Assay (MDA Levels)

MDA levels, an indicator of lipid peroxidation, were determined using the thiobarbituric acid (TBA) assay. The reaction mixture included 50 µL of 8.1% sodium dodecyl sulfate (SDS), 150 µL of distilled water, 375 µL of 0.8% TBA, 375 µL of 20% acetic acid, and 50 µL of tissue homogenate. Samples were incubated at 95 °C for 60 min, then cooled and mixed with 250 µL of distilled water and 1250 µL of n-butanol–pyridine (15:1, *v*/*v*). Following centrifugation at 4000 rpm for 10 min, the absorbance of the upper organic layer was measured at 532 nm. MDA concentrations were quantified using a calibration curve prepared from 1,1,3,3-tetramethoxypropane (TMP) and reported as ng/mg protein [53].

### 4.6. Western Blotting

Cardiac and vascular tissues were lysed using RIPA buffer containing 20 mM Tris-HCl (pH 7.5), 150 mM NaCl, 1 mM Na_2_EDTA, 1 mM EGTA, 1% NP-40, 1% sodium deoxycholate, 2.5 mM sodium pyrophosphate, 1 mM β-glycerophosphate, 1 mM Na_3_VO_4_, 1 µg/mL leupeptin, and 1 mM PMSF (Sigma-Aldrich). This buffer composition was optimized to preserve protein structure and inhibit enzymatic degradation during extraction. Following tissue homogenization, samples were centrifuged at 12,000× *g* for 10 min at 4 °C. The resulting supernatant, representing the total protein fraction, was collected for further analysis. Protein content was quantified by measuring absorbance at 280 nm using a NanoDrop 2000c spectrophotometer (Thermo Fisher Scientific).

Protein lysates (60 µg) were mixed with SDS-PAGE loading buffer containing SDS and a reducing agent, then heated at 95 °C for 10 min to ensure complete denaturation. The prepared samples were separated on a 10% SDS-PAGE gel based on molecular weight. Following electrophoresis, proteins were transferred onto nitrocellulose membranes using a wet transfer system at 100 V for one hour.

The membrane was rinsed with TBS-T (0.05% Tween 20 in Tris-buffered saline) to eliminate residual transfer buffer. Non-specific binding sites were blocked by incubating the membrane in 5% skim milk prepared in 0.1% TBS-T for one hour at room temperature. Subsequently, the membrane was incubated overnight at 4 °C with the following primary antibodies (all from Cell Signaling Technology, Danvers, MA, USA; 1:1000 dilution): anti-sirtuin, anti-NF-κB, anti-IL-6, anti-TNF-α, anti-TGF-β, and anti-β-actin. These antibodies specifically bind to their target proteins for subsequent detection.

After primary antibody incubation, the membrane was washed three times with TBS-T to remove unbound antibodies. It was then incubated for one hour at room temperature with an HRP-conjugated anti-rabbit secondary antibody (Cell Signaling Technology; 1:2000 dilution), which enables chemiluminescent detection by binding to the primary antibodies. Excess secondary antibody was removed by additional washes with TBS-T.

The Clarity™ Western Enhanced Chemiluminescence (ECL) Substrate (Bio-Rad Laboratories, Inc., Hercules, CA, USA) was used to visualize the protein bands, and an LAS-4000 luminescent image analyzer (GE Healthcare, Chicago, IL, USA) was used to capture the chemiluminescent signal. Protein band intensity was measured using GE Healthcare’s ImageQuant TL v.7.0 image analysis software (GE Healthcare). Band densities were standardized to the anti-β-actin loading control to adjust for differences in protein loading. Data were presented as relative band density in comparison to the naïve control group [54,55]. The full-length membranes with visible membrane edges are shown in the Appendix A for SIRT1 expression in cardiac tissue, Appendix A for SIRT1 expression in vascular tissue, Appendix A for NF-κB expression in cardiac tissue, Appendix A for NF-κB expression in vascular tissue, Appendix A for TNF-α expression in cardiac tissue, Appendix A for TNF-α expression in vascular tissue, Appendix A for IL-6 expression in cardiac tissue, and Appendix A for TGF-β expression in vascular tissue.

### 4.7. Analysis of Statistics

Data are presented as mean ± standard error of the mean (SEM). Statistical significance was determined using one-way analysis of variance (ANOVA), followed by Tukey’s post hoc test for pairwise comparisons. A *p*-value of less than 0.05 was considered statistically significant. All analyses were conducted using SPSS software, version 29.0.2 (IBM Corp., Armonk, NY, USA).

## 5. Conclusions

This study demonstrates that PZT confers significant cardiovascular protection against PM2.5-induced systemic inflammation through SIRT1-mediated epigenetic modulation. PZT pretreatment attenuated oxidative stress, suppressed pro-inflammatory cytokine expression, and restored antioxidant enzyme activities in both cardiac and vascular tissues. Notably, the upregulation of SIRT1 underscores its central role in orchestrating the anti-inflammatory and antioxidant responses elicited by PZT, as shown in Figure 10. These protective effects were dose-dependent, with higher doses (200 and 400 mg/kg BW) yielding more pronounced benefits. Overall, PZT shows promise as both a preventive and therapeutic agent by modulating SIRT1-related pathways to mitigate pollution-induced cardiovascular damage. Further research is needed to confirm its long-term effectiveness and explore potential clinical applications.

## Figures and Tables

**Figure 1 ijms-26-05759-f001:**
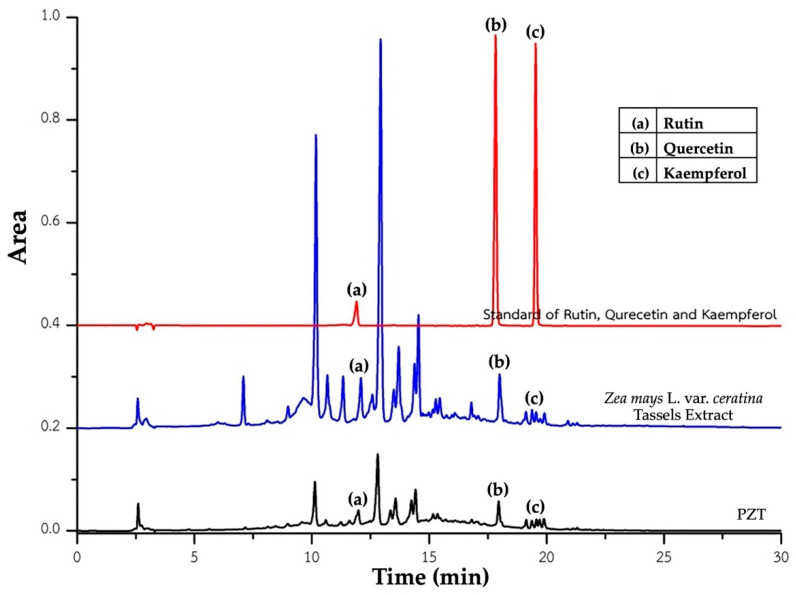
Chromatographic fingerprint of rutin, quercetin, and kaempferol in *Zea mays* L. var. *ceratina* tassel extract and its phytosome formulation (PZT). PZT refers to phytosome-encapsulated *Zea mays* L. var. *ceratina* tassel extract.

**Figure 2 ijms-26-05759-f002:**
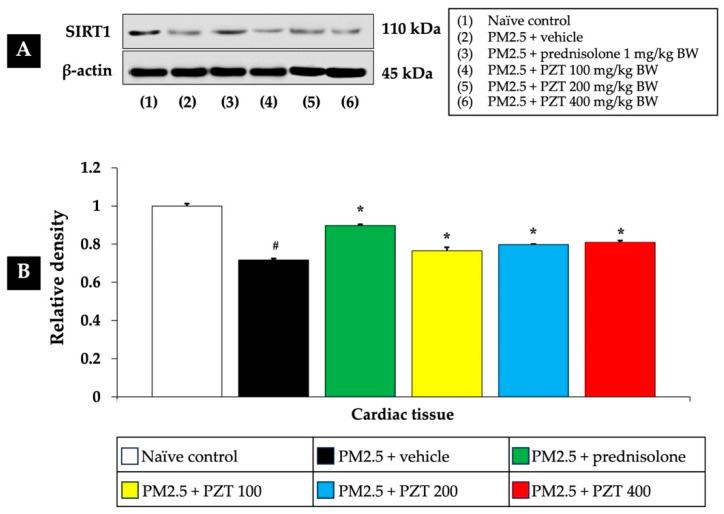
Effects of PZT on SIRT1 expression in cardiac tissue. (**A**) Representative Western blot image showing the expression levels of SIRT1 protein. (**B**) Bar graph presenting the relative protein expression of SIRT1, normalized to β-actin. Data are expressed as mean ± SEM. ^#^
*p* < 0.05, compared to the naïve control group. * *p* < 0.05, compared to the PM2.5-exposed+vehicle group. Prednisolone refers to treatment with prednisolone at 1 mg/kg BW; PZT100, PZT200, and PZT400 refer to treatment with phytosome-encapsulated *Zea mays* L. var. *ceratina* tassel extract at 100, 200, and 400 mg/kg BW, respectively; and SIRT1 refers to sirtuin 1.

**Figure 3 ijms-26-05759-f003:**
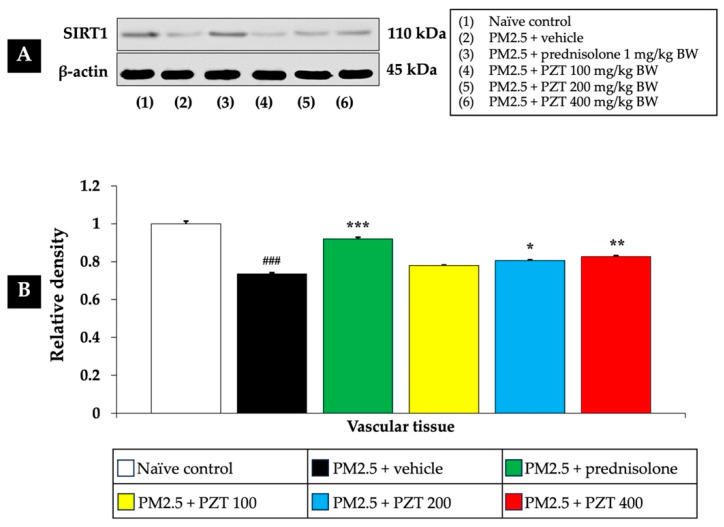
Effects of PZT on SIRT1 expression in vascular tissue. (**A**) Representative Western blot image showing the expression levels of SIRT1 protein. (**B**) Bar graph presenting the relative protein expression of SIRT1, normalized to β-actin. Data are expressed as mean ± SEM. ^###^
*p* < 0.001, compared to the naïve control group. *, **, and *** indicate *p* < 0.05, 0.01, and 0.001, respectively, compared to the PM2.5-exposed+vehicle group. Prednisolone refers to treatment with prednisolone at 1 mg/kg BW; PZT100, PZT200, and PZT400 refer to treatment with phytosome-encapsulated *Zea mays* L. var. *ceratina* tassel extract at 100, 200, and 400 mg/kg BW, respectively; and SIRT1 refers to sirtuin 1.

**Figure 4 ijms-26-05759-f004:**
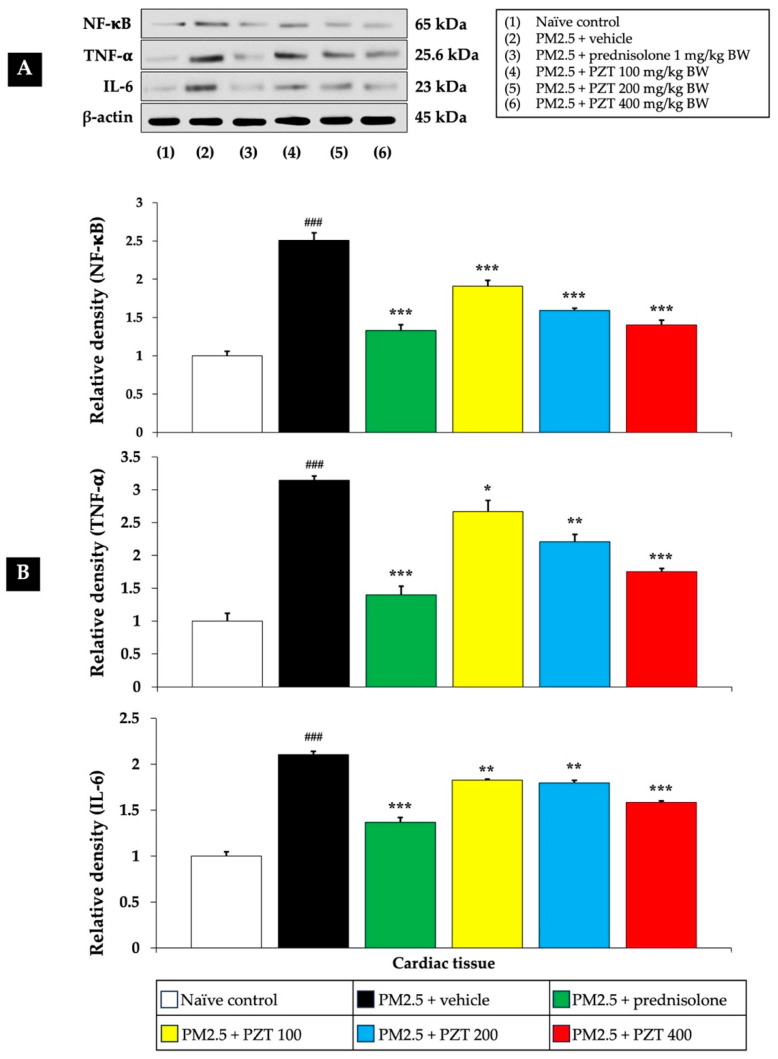
Effects of PZT on inflammatory markers, including NF-κB, TNF-α, and IL-6 expressions in cardiac tissue. (**A**) Representative Western blot image showing the expression levels of NF-κB, TNF-α, and IL-6 protein. (**B**) Bar graph presenting the relative protein expression of NF-κB, TNF-α, and IL-6, normalized to β-actin. Data are expressed as mean ± SEM. ^###^
*p* < 0.001, compared to the naïve control group. *, **, and *** indicate *p* < 0.05, 0.01, and 0.001, respectively, compared to the PM2.5-exposed+vehicle group. Prednisolone refers to treatment with prednisolone at 1 mg/kg BW; PZT100, PZT200, and PZT400 refer to treatment with phytosome-encapsulated *Zea mays* L. var. *ceratina* tassel extract at 100, 200, and 400 mg/kg BW, respectively; NF-κB refers to nuclear factor-kappa B; TNF-α refers to tumor necrosis factor-alpha; and IL-6 refers to interleukin-6.

**Figure 5 ijms-26-05759-f005:**
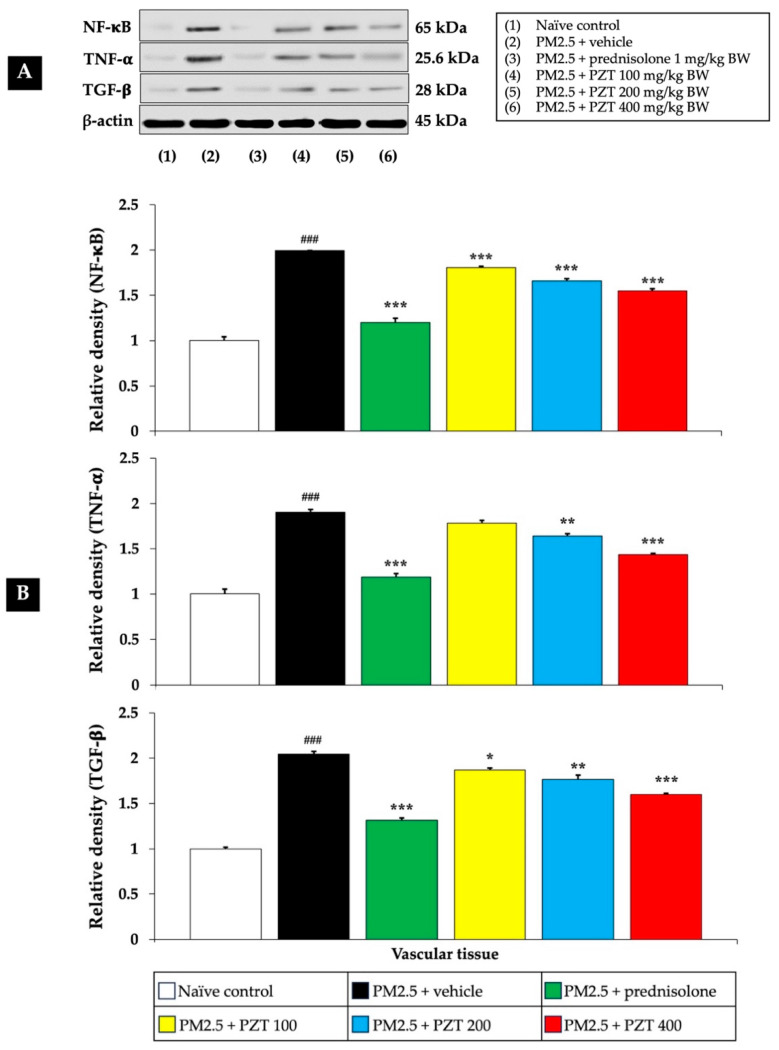
Effects of PZT on inflammatory markers, including NF-κB, TNF-α, and TGF-β expressions in vascular tissue. (**A**) Representative Western blot image showing the expression levels of NF-κB, TNF-α, and TGF-β protein. (**B**) Bar graph presenting the relative protein expression of NF-κB, TNF-α, and TGF-β, normalized to β-actin. Data are expressed as mean ± SEM. ^###^
*p* < 0.001, compared to the naïve control group. *, **, and *** indicate *p* < 0.05, 0.01, and 0.001, respectively, compared to the PM2.5-exposed+vehicle group. Prednisolone refers to treatment with prednisolone at 1 mg/kg BW; PZT100, PZT200, and PZT400 refer to treatment with phytosome-encapsulated *Zea mays* L. var. *ceratina* tassel extract at 100, 200, and 400 mg/kg BW, respectively; NF-κB refers to nuclear factor-kappa B; TNF-α refers to tumor necrosis factor-alpha; and TGF-β refers to transforming growth factor-beta.

**Figure 6 ijms-26-05759-f006:**
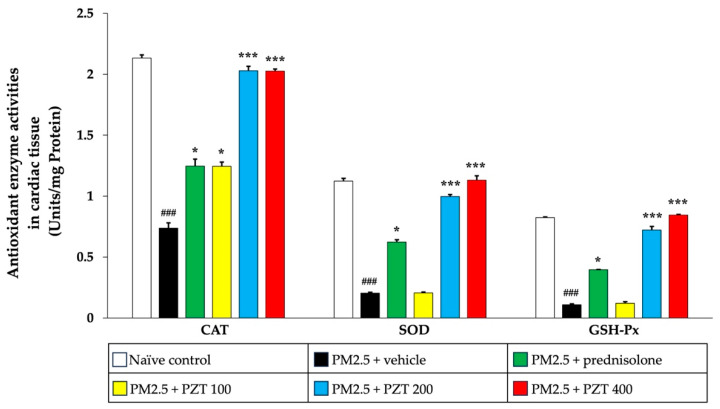
Effects of PZT on antioxidant enzyme activities in cardiac tissue. Data are presented as mean ± SEM. ^###^
*p* < 0.001, compared to the naïve control group. * and *** indicate *p* < 0.05 and 0.001, respectively, compared to the PM2.5-exposed vehicle group. Prednisolone refers to treatment with prednisolone at 1 mg/kg BW; PZT100, PZT200, and PZT400 refer to treatment with phytosome-encapsulated *Zea mays* L. var. *ceratina* tassel extract at 100, 200, and 400 mg/kg BW, respectively; CAT refers to catalase; SOD refers to superoxide dismutase; and GSH-Px refers to glutathione peroxidase.

**Figure 7 ijms-26-05759-f007:**
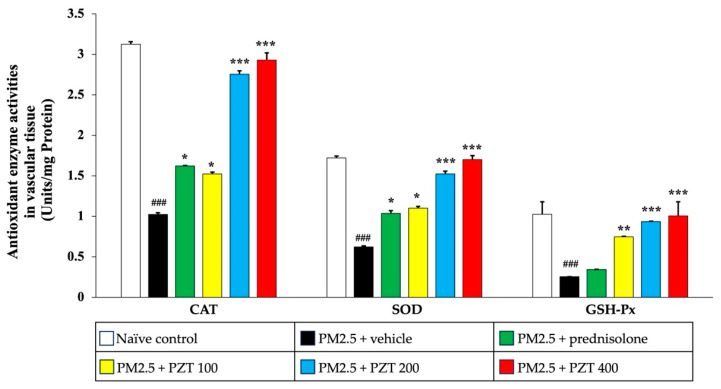
Effects of PZT on antioxidant enzyme activities in vascular tissue. Data are presented as mean ± SEM. ^###^ *p* < 0.001, compared to the naïve control group. *, ** and *** indicate *p* < 0.05, 0.01, and 0.001, respectively, compared to the PM2.5-exposed vehicle group. Prednisolone refers to treatment with prednisolone at 1 mg/kg BW; PZT100, PZT200, and PZT400 refer to treatment with phytosome-encapsulated *Zea mays* L. var. *ceratina* tassel extract at 100, 200, and 400 mg/kg BW, respectively; CAT refers to catalase; SOD refers to superoxide dismutase; and GSH-Px refers to glutathione peroxidase.

**Figure 8 ijms-26-05759-f008:**
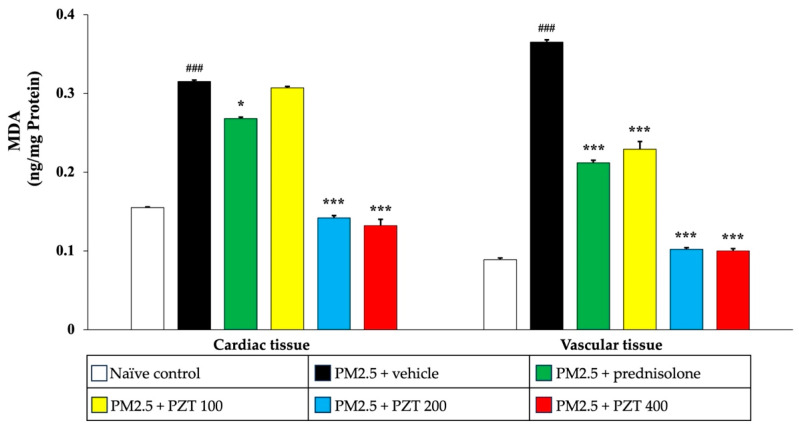
Effects of PZT on malondialdehyde levels in cardiac and vascular tissues. Data are presented as mean ± SEM. ^###^ *p* < 0.001, compared to the naïve control group. * and *** indicate *p* < 0.05 and 0.001, respectively, compared to the PM2.5-exposed vehicle group. Prednisolone refers to treatment with prednisolone at 1 mg/kg BW; PZT100, PZT200, and PZT400 refer to treatment with phytosome-encapsulated *Zea mays* L. var. *ceratina* tassel extract at 100, 200, and 400 mg/kg BW, respectively; and MDA refers to malondialdehyde.

**Figure 9 ijms-26-05759-f009:**
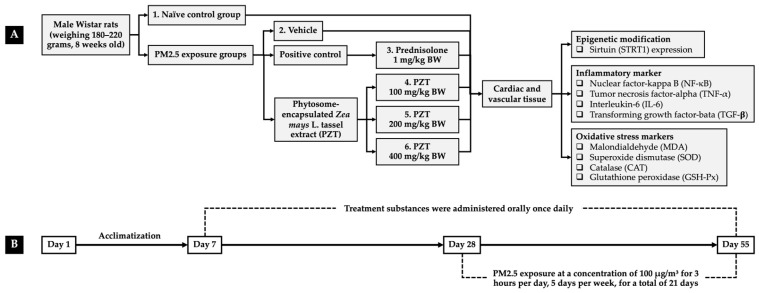
Schematic representation of the experimental procedures. (**A**) Overview of the protocol detailing PZT administration and the assessment of key parameters. (**B**) Timeline depicting the induction of PM2.5-induced systemic inflammation and the corresponding schedule for PZT treatment. PZT refers to phytosome-encapsulated *Zea mays* L. var. *ceratina* tassel extract.

**Figure 10 ijms-26-05759-f010:**
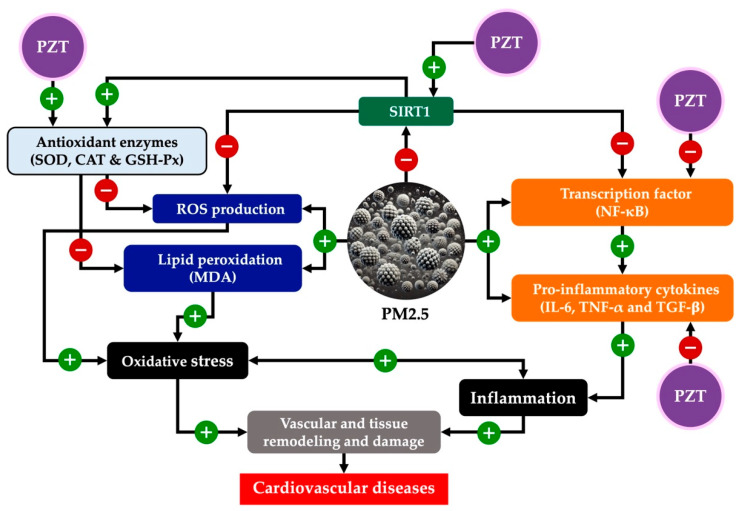
Schematic diagram summarizing the pathophysiology of PM2.5-induced cardiovascular injury and the potential mechanisms underlying PZT treatment. PZT refers to phytosome-encapsulated *Zea mays* L. var. *ceratina* tassel extract; PM2.5 refers to fine particulate matter <2.5 µm; NF-κB refers to nuclear factor kappa B; IL-6 refers to interleukin-6; TNF-α refers to tumor necrosis factor-alpha; TGF-β: refers to transforming growth factor-beta; ROS refers to reactive oxygen species; SOD refers to superoxide dismutase; CAT refers to catalase; GSH-Px refers to glutathione peroxidase; and MDA: refers to malondialdehyde.

**Table 1 ijms-26-05759-t001:** Active compounds of *Zea mays* L. var. *ceratina* tassel extract and PZT.

Active Compounds	Composition	*Zea mays* L. var. *ceratina* Tassel Extract (µg/mg)	PZT (µg/mg)	Linearity Range (μg/mL)	Linear Equation	Coefficient of Determination (r^2^)
Rutin	C_27_H_30_O_16_	0.730 ± 0.012	0.602 ± 0.022	0.25–10	y = 16,315x − 5132.6	0.9977
Quercetin	C_15_H_10_O_7_	0.077 ± 0.003	0.078 ± 0.006	0.25–10	y = 187,644x − 55,225	0.9981
Kaempferol	C_15_H_10_O_6_	0.042 ± 0.000	0.074 ± 0.000	0.05–40	y = 124,324x − 176,927	0.9906

Data are expressed as mean ± SEM.

## Data Availability

The original contributions presented in this study are included in the article and Appendix A. Further inquiries can be directed to the corresponding authors.

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
