# Peer review of "SIRT1-Mediated Epigenetic Protective Mechanisms of Phytosome-Encapsulated Zea mays L. var. ceratina Tassel Extract in a Rat Model of PM2.5-Induced Cardiovascular Inflammation"

_ijms, 2025, doi:10.3390/ijms26125759_

Round 1

Reviewer 1 Report

Comments and Suggestions for Authors

This manuscript discusses the therapeutic impact of phytosome-encapsulated Zea mays L. var. ceratina tassel extract (PZT) on cardiovascular injury induced by fine particulate matter (PM2.5) in rats. In brief, this study addresses a key public health issue, cardiovascular injury from air pollution, by targeting underlying epigenetic mechanisms, specifically SIRT1, and not just symptoms.
I have some questions for authors:
1. Why are only male Wistar rats used by the author in this study? These results may not directly translate to humans, and there were no female animals were used, without potential sex-based differences.
2. Second, the duration of PM2.5 exposure and PZT treatment (27 and 21 days, respectively), may not fully capture the chronic effects relevant to human cardiovascular disease.
3. While the phytosome’s effect on flavonoid content is reported, there is no direct measurement of systemic bioavailability or pharmacokinetics in vivo.
4. I am also questioning the mechanistic specificity here; this study attributes effects primarily to SIRT1 but does not explore other sirtuins or epigenetic regulators that may also play roles.
5. This study focuses on molecular and biochemical markers but lacks direct measurement of cardiovascular function (e.g., blood pressure, echocardiography).
6. Some background information is repeated in this paper, and the introduction could be concise.
Comments for authors:
1. The author must include male and female animals to be able to assess sex-specific effects.
2. The author must extend the study duration or include a chronic exposure paradigm for greater translational relevance.
3. Directly assess the pharmacokinetics and systemic bioavailability of key bioactives after phytosome encapsulation.
4. Examine additional epigenetic regulators and downstream targets to improve mechanistic conclusions.
5. Incorporate functional cardiovascular assessments to complement molecular findings.

Author Response

Response to reviewer and editor suggestion

We sincerely thank you for your letter and the reviewers’ insightful comments on our manuscript, SIRT1-Mediated Epigenetic Protective Mechanisms of Phytosome-Encapsulated Zea mays L. var. ceratina Tassel Extract in a Rat Model of PM2.5-Induced Cardiovascular Inflammation (Manuscript ID: ijms-3668623).

We greatly appreciate the opportunity to revise our manuscript and are grateful for the constructive feedback. We apologize for any errors in the initial submission and acknowledge the reviewers’ invaluable input, which has helped enhance the scientific rigor and clarity of our work.

We have carefully considered each comment and made revisions accordingly. Below, we provide a detailed account of the main corrections and our responses to the reviewers’ suggestions.

Response to reviewer 1
This manuscript discusses the therapeutic impact of phytosome-encapsulated Zea mays L. var. ceratina tassel extract (PZT) on cardiovascular injury induced by fine particulate matter (PM2.5) in rats. In brief, this study addresses a key public health issue, cardiovascular injury from air pollution, by targeting underlying epigenetic mechanisms, specifically SIRT1, and not just symptoms.

I have some questions for authors:

Question 1: Why are only male Wistar rats used by the author in this study? These results may not directly translate to humans, and there were no female animals were used, without potential sex-based differences.

Response to question 1: Thank you for raising this important point. In this study, we used only male Wistar rats to minimize the influence of hormonal fluctuations associated with the estrous cycle in females, which can introduce variability in inflammatory, oxidative stress, and epigenetic parameters. This approach is commonly adopted in preclinical studies to maintain consistency and reduce biological variability during the initial phases of mechanistic investigation.

We fully acknowledge the importance of considering sex-based differences, especially in translational research. However, the current study was designed as an initial proof-of-concept to explore the protective mechanisms of the phytosome formulation (PZT). Future studies will incorporate both male and female animals to evaluate potential sex-dependent responses and enhance the translational relevance of the findings.

Question 2: Second, the duration of PM2.5 exposure and PZT treatment (27 and 21 days, respectively), may not fully capture the chronic effects relevant to human cardiovascular disease.

Response to question 2: Thank you for your insightful comment. We agree that chronic exposure models are highly relevant when studying cardiovascular outcomes linked to PM2.5. In our study, rats were exposed to PM2.5 aerosol for 3 hours per day, five days per week, over a period of 21 days. This exposure regimen was carefully designed to mimic environmental and occupational conditions, simulating a chronic, subacute exposure pattern representative of real-world scenarios.

Importantly, a two-day break each week allowed for a short recovery period, reflecting the intermittent nature of real human exposure, while the daily 3-hour exposure provided a sufficient PM2.5 dose to induce systemic oxidative stress and inflammation without triggering acute toxicity. The 21-day duration aligns with previous models demonstrating reliable pathophysiological changes in respiratory and cardiovascular systems within a manageable experimental window.

Reference:

  • Sivakumar, B.; Ali, N.; Ahmad, S.F.; Nadeem, A.; Waseem, M.; Kurian, G.A. PM5-Induced Cardiac Structural Modifications and Declined Pro-Survival Signalling Pathways Are Responsible for the Inefficiency of GSK-3β Inhibitor in Attenuating Myocardial Ischemia-Reperfusion Injury in Rats. Cells202312, 2064. https://doi.org/10.3390/cells12162064

Moreover, our previous publication using this same exposure protocol has shown that this timeframe is sufficient to induce significant lung pathology, including inflammation, oxidative stress, and histopathological alterations in a PM2.5-induced lung injury model

Reference:

  • Palachai, N.; Thukham-mee, W.; Wattanathorn, J. The Protective Effect against Lung Injury of Phytosome Containing the Extract of Purple Waxy Corn Tassel in an Animal Model of PM2.5-Induced Lung Inflammation. Foods 2024, 13, 3258. https://doi.org/10.3390/foods13203258

It is also worth noting that rats have a significantly shorter lifespan compared to humans (approximately 2–3 years), and therefore, a 21-day exposure in rats represents a proportionally longer duration in the context of their lifespan. This period is suitable for capturing early but relevant systemic effects of PM2.5, including inflammatory responses and oxidative damage, which are key contributors to cardiovascular risk.

While our current focus was on early and subacute effects, we agree that longer-term studies are needed to better understand chronic disease progression. We intend to expand upon this work in future experiments using extended exposure durations to further evaluate the long-term efficacy and protective mechanisms of PZT.

Question 3: While the phytosome’s effect on flavonoid content is reported, there is no direct measurement of systemic bioavailability or pharmacokinetics in vivo.

Response to question 3: Thank you for this important observation. We acknowledge that our study did not include direct in vivo measurements of systemic bioavailability or pharmacokinetics of the flavonoid compounds delivered through the phytosome formulation. Our primary objective in this study was to evaluate the protective effects of the PZT extract against PM2.5-induced lung and systemic inflammation.

However, the enhancement of flavonoid content and improved efficacy observed in the PZT group compared to the non-formulated extract group indirectly supports the improved bioavailability conferred by the phytosome delivery system. Phytosomes are known to enhance the absorption and systemic availability of flavonoids and other polyphenols, as supported by prior literature, owing to their improved solubility and permeability through biological membranes.

While the pharmacokinetic behavior of PZT components was not measured in this study, this remains a priority for our future work. We plan to conduct detailed pharmacokinetic analyses and tissue distribution studies in upcoming experiments to better quantify the bioavailability and in vivo behavior of the key flavonoids (such as cyanidin-3-glucoside, rutin, quercetin, and kaempferol) in the phytosome form.

We have now added a note in the discussion to acknowledge this limitation and highlight the need for future investigation.

Question 4: I am also questioning the mechanistic specificity here; this study attributes effects primarily to SIRT1 but does not explore other sirtuins or epigenetic regulators that may also play roles.

Response to question 4: Thank you for raising this insightful point. We agree that epigenetic regulation involves a complex network of mediators, including not only SIRT1 but also other sirtuins (e.g., SIRT3, SIRT6) and various histone-modifying enzymes, DNA methyltransferases, and non-coding RNAs that may contribute to the observed protective effects.

In this study, we specifically focused on SIRT1 due to its well-established role in regulating oxidative stress, inflammation, and mitochondrial function—processes central to PM2.5-induced toxicity. SIRT1 has also been identified as a key target for several flavonoids, including quercetin and anthocyanins, which are abundant in the PZT extract. Our findings of increased SIRT1 expression in the PZT-treated group support its involvement in the protective mechanism.

References:

  1. Lai, T., Wen, X., Wu, D., Su, G., Gao, Y., Chen, C., Wu, W., Lv, Y., Chen, Z., Lv, Q., Li, W., Li, D., Chen, M., & Wu, B. (2019). SIRT1 protects against urban particulate matter-induced airway inflammation. International journal of chronic obstructive pulmonary disease14, 1741–1752. https://doi.org/10.2147/COPD.S202904
  2. Yan, Q., Zheng, R., Li, Y., Hu, J., Gong, M., Lin, M., Xu, X., Wu, J., & Sun, S. (2024). PM5-induced premature senescence in HUVECs through the SIRT1/PGC-1α/SIRT3 pathway. The Science of the total environment921, 171177. https://doi.org/10.1016/j.scitotenv.2024.171177
  3. Chung, S., Yao, H., Caito, S., Hwang, J. W., Arunachalam, G., & Rahman, I. (2010). Regulation of SIRT1 in cellular functions: role of polyphenols. Archives of biochemistry and biophysics501(1), 79–90. https://doi.org/10.1016/j.abb.2010.05.003
  4. Peng, J., Yang, Z., Li, H., Hao, B., Cui, D., Shang, R., Lv, Y., Liu, Y., Pu, W., Zhang, H., He, J., Wang, X., & Wang, S. (2023). Quercetin Reprograms Immunometabolism of Macrophages via the SIRT1/PGC-1α Signaling Pathway to Ameliorate Lipopolysaccharide-Induced Oxidative Damage. International journal of molecular sciences24(6), 5542. https://doi.org/10.3390/ijms24065542

Nonetheless, we fully acknowledge that other sirtuins and epigenetic pathways may also contribute. Our current focus on SIRT1 was intended as an initial step in unraveling the epigenetic basis of PZT’s effects. We are actively planning future studies to investigate a broader spectrum of sirtuins (e.g., SIRT3 and SIRT6) and other epigenetic regulators such as histone acetyltransferases (HATs), histone deacetylases (HDACs), and non-coding RNAs, in order to provide a more comprehensive mechanistic understanding.

We have included this limitation in the revised discussion and emphasized the need for broader epigenetic profiling in future investigations.

Question 5: This study focuses on molecular and biochemical markers but lacks direct measurement of cardiovascular function (e.g., blood pressure, echocardiography).

Response to question 5: Thank you for your valuable comment. We agree that assessing cardiovascular function through parameters such as blood pressure, echocardiography, or vascular reactivity would provide more comprehensive insight into the functional impact of PM2.5 exposure and the protective effects of PZT.

In this study, our primary objective was to investigate the early pathophysiological changes at the molecular and biochemical levels—particularly oxidative stress, inflammatory markers, and epigenetic regulation via SIRT1—as these are early drivers and predictive indicators of cardiovascular pathology. The use of these biomarkers is consistent with prior studies investigating subacute or early-stage responses to environmental toxins and therapeutic interventions.

Moreover, this study was designed as a foundational step to explore the mechanistic underpinnings of PZT’s protective effects. The findings lay essential groundwork for subsequent studies incorporating direct assessments of cardiovascular function. We have acknowledged this limitation in the revised manuscript and emphasized the need for future investigations involving functional endpoints such as arterial blood pressure monitoring, vascular ultrasound, or echocardiographic evaluations to establish direct correlations with physiological outcomes.

We appreciate the reviewer’s suggestion and agree it is a crucial direction for expanding the translational relevance of our findings.

Question 6: Some background information is repeated in this paper, and the introduction could be concise.

Response to question 6: We have carefully revised the introduction to improve conciseness and reduce repetitive background information. The revised introduction now presents a more focused and streamlined overview of cardiovascular diseases, the impact of PM2.5, the role of SIRT1 in epigenetic regulation, and the potential of PZT. These changes enhance clarity while maintaining the necessary context and rationale for the study.

Comments for authors:

Comment 1:  The author must include male and female animals to be able to assess sex-specific effects.

Response to comment 1: Thank you for your comment. As noted earlier, we used only male Wistar rats in this initial study to reduce variability related to hormonal fluctuations in females, which can affect molecular and inflammatory outcomes. This approach is common in early mechanistic studies to enable clearer interpretation of primary endpoints.

            Nonetheless, we recognize that sex-based differences play a significant role in both disease susceptibility and therapeutic responses. We have noted this limitation in the revised manuscript and agree that future studies should include both sexes to assess differential responses to PM2.5 exposure and PZT treatment. Such studies are planned as a follow-up to this work, with the aim of expanding the generalizability and translational relevance of our findings.

Comment 2:  The author must extend the study duration or include a chronic exposure paradigm for greater translational relevance.

Response to comment 2: Thank you for your valuable comment. We agree that extended or chronic exposure models would enhance translational relevance. In this initial study, we aimed to investigate early pathophysiological changes using a subacute exposure model. As noted in the discussion, future studies are planned to assess the long-term effects and therapeutic potential of PZT under chronic PM2.5 exposure conditions.

Comment 3:  Directly assess the pharmacokinetics and systemic bioavailability of key bioactives after phytosome encapsulation.

Response to comment 3: Thank you for your insightful comment. We acknowledge the importance of directly assessing pharmacokinetics and systemic bioavailability. Although this was beyond the scope of the current study, we have highlighted it as a key limitation and plan to address it in future investigations to strengthen the translational potential of PZT.

Comment 4:  Examine additional epigenetic regulators and downstream targets to improve mechanistic conclusions.

Response to comment 4: Thank you for your constructive suggestion. We focused on SIRT1 as a key epigenetic regulator due to its well-established role in modulating oxidative stress and inflammation. However, we agree that investigating additional epigenetic regulators and downstream targets would strengthen the mechanistic conclusions. This limitation has been acknowledged in the discussion, and future studies are planned to expand on these molecular pathways.

Comment 5:  Incorporate functional cardiovascular assessments to complement molecular findings.

Response to comment 5: Thank you for your helpful comment. We agree that incorporating functional cardiovascular assessments, such as blood pressure or echocardiography, would enhance the translational impact of our findings. This limitation has been acknowledged in the discussion, and we plan to include these assessments in future studies.

“In summary, our results demonstrate that PZT mitigates PM2.5-induced cardio-vascular injury via a multi-targeted mechanism involving oxidative stress reduction, inflammation suppression, and epigenetic modulation through SIRT1 activation. How-ever, several limitations should be acknowledged. First, this study used only male rats to minimize biological variability; future studies should include both sexes to investigate sex-specific responses. Second, while the 21-day PM2.5 exposure regimen simulates subacute real-world exposure and aligns with previous models, longer durations are needed to better capture chronic pathological effects. Third, although we focused on SIRT1 due to its well-established relevance to oxidative and inflammatory pathways and its interaction with dietary flavonoids, future studies should explore other sirtuins and epigenetic regulators. Fourth, the study did not include pharmacokinetic or systemic bioavailability assessments of the key bioactive compounds, which are critical for con-firming enhanced delivery and informing dose translation. Lastly, the lack of direct cardiovascular functional measurements (e.g., blood pressure or echocardiography) limits the translation of molecular findings into clinical relevance; future studies should in-corporate such evaluations.

Taken together, our findings provide a strong mechanistic basis for the protective role of PZT in PM2.5-induced injury and offer promising implications for its use as a pre-ventive or therapeutic strategy in populations exposed to environmental air pollutants. Future investigations should prioritize pharmacokinetic profiling, sex-specific analyses, functional cardiovascular endpoints, and extended exposure durations to support trans-lational and clinical development.”

Thank you once again for your valuable feedback. We appreciate the time and effort invested by the reviewers and editor in evaluating our manuscript. We have carefully addressed each point raised and made necessary revisions accordingly. We eagerly await further feedback and guidance from the editorial team.

Yours sincerely,

Nut Palachai

Reviewer 2 Report

Comments and Suggestions for Authors

General comments:  

Very interesting study but with some minor corrections it can be considered for publication.

Title:

“SIRT1-Mediated Epigenetic Modulation by Phytosome-Encapsulated Zea mays L. var. ceratina Tassel Extract Confers Cardiovascular Protection in PM2.5-Induced Systemic Inflammation” – This title needs to be reworded for clarity.

Consider rewording the title to something like this: “SIRT -mediated epigenetic protective mechanisms of Phytosome-Encapsulated Zea mays L. var. ceratina Tassel Extract in a rat model of PM2.5-Induced cardiovascular Inflammation “

Abstract:

Lines 31-32 – “functional food therapeutic”? The experiment used a preventative approach rather than a therapeutic approach.

Introduction:

Lines 64-65 – insert references to explain the poor bioavailability and stability. There are studies on the bioavailability of grain phenolics. See this example doi: 10.3390/foods10071595

Lines 69-71 – why was phytosome used for encapsulation?

Line 70 – 73 what control was used?

Methods:   

Line 549 – what is the vehicle? Please clarify early on in the methods section.

Results:

Define abbreviations used in all figures and tables. For example Figure 6 and 7 should have CAT, SOD and GSH-Px abbreviated

Line 89-90 Why were the phenolic compounds selected were not mentioned din the introduction?

Discussion

Line 229 -230 The first sentence of the summary should focus on the effects of PZT and not PM2.5

How does this study compare to human trials if any?

Author Response

Response to reviewer and editor suggestion

We sincerely thank you for your letter and the reviewers’ insightful comments on our manuscript, SIRT1-Mediated Epigenetic Protective Mechanisms of Phytosome-Encapsulated Zea mays L. var. ceratina Tassel Extract in a Rat Model of PM2.5-Induced Cardiovascular Inflammation (Manuscript ID: ijms-3668623).

We greatly appreciate the opportunity to revise our manuscript and are grateful for the constructive feedback. We apologize for any errors in the initial submission and acknowledge the reviewers’ invaluable input, which has helped enhance the scientific rigor and clarity of our work.

We have carefully considered each comment and made revisions accordingly. Below, we provide a detailed account of the main corrections and our responses to the reviewers’ suggestions.

Response to reviewer 2

Comments and Suggestions for Authors

General comments: Very interesting study but with some minor corrections it can be considered for publication.

Response: Thank you for your positive feedback and for considering our study interesting. We appreciate your constructive review and have carefully addressed all the minor corrections as suggested. We believe these revisions have improved the clarity and quality of the manuscript, and we hope it now meets the standards for publication.

Title:

Comment 1:  “SIRT1-Mediated Epigenetic Modulation by Phytosome-Encapsulated Zea mays L. var. ceratina Tassel Extract Confers Cardiovascular Protection in PM2.5-Induced Systemic Inflammation” – This title needs to be reworded for clarity.

Consider rewording the title to something like this: “SIRT -mediated epigenetic protective mechanisms of Phytosome-Encapsulated Zea mays L. var. ceratina Tassel Extract in a rat model of PM2.5-Induced cardiovascular Inflammation “

Response to comment 1: We appreciate the reviewer’s valuable suggestion regarding the clarity of the manuscript title. In response, we have revised the title to better reflect the study focus and enhance clarity, as recommended. The revised title is:

SIRT1-Mediated Epigenetic Protective Mechanisms of Phytosome-Encapsulated Zea mays L. var. ceratina Tassel Extract in a Rat Model of PM2.5-Induced Cardiovascular Inflammation

We believe this revised version more accurately communicates the study’s key components and mechanisms while maintaining scientific precision. Thank you for the constructive feedback.

Abstract:

Comment 2:  Lines 31-32 – “functional food therapeutic”? The experiment used a preventative approach rather than a therapeutic approach.

Response to comment 2: We thank the reviewer for the insightful observation. We agree that the phrasing “functional food-based therapeutic” may be misleading, as the study design employed a preventative rather than a therapeutic approach. In response, we have revised the final sentence of the abstract to more accurately reflect the preventative nature of the intervention. The revised sentence now reads:

“The dual anti-inflammatory and antioxidant actions of PZT via SIRT1 activation highlight its potential as a functional food-based preventative agent for reducing cardiovascular risk in polluted environments.”

We appreciate the reviewer’s valuable input, which has helped improve the clarity and accuracy of our abstract.

Introduction:

Comment 3:  Lines 64-65 – insert references to explain the poor bioavailability and stability. There are studies on the bioavailability of grain phenolics. See this example doi: 10.3390/foods10071595

Response to comment 3: We thank the reviewer for the helpful suggestion. In response, we have added appropriate references to support the statement regarding the poor bioavailability and stability of phenolic compounds in grain-based materials. Specifically, we have cited the recommended article:

Ed Nignpense, B.; Francis, N.; Blanchard, C.; Santhakumar, A.B. Bioaccessibility and Bioactivity of Cereal Polyphenols: A Review. Foods 202110, 1595. https://doi.org/10.3390/foods10071595

This reference has been incorporated to strengthen the rationale for using a phytosome-based delivery system to enhance bioavailability.

Comment 4:  Lines 69-71 – why was phytosome used for encapsulation?

Response to comment 4: We appreciate the reviewer’s thoughtful comment. To clarify the rationale for selecting phytosome encapsulation, we have revised the text to emphasize not only its benefits in enhancing solubility and bioavailability but also its use of phosphatidylcholine—a natural cell membrane component—which facilitates improved cellular uptake and targeted delivery.

Revised sentence in the Introduction:

To overcome these limitations, phytosome encapsulation—a delivery system that complexes bioactive compounds with phospholipids, particularly phosphatidylcholine—has been employed. This approach not only enhances solubility, stability, and intestinal absorption but also facilitates efficient cellular uptake, as phosphatidylcholine is a key component of cell membranes and enables easier passage through lipid bilayers [10–13].

Comment 5:  Line 70 – 73 what control was used?

Response to comment 5: We appreciate the reviewer’s comment. The experimental design, including the details of the control group used in the study, is thoroughly described in the Materials and Methods section. We respectfully refer the reviewer to that section for a full explanation of the control conditions implemented.

Methods:  

Comment 6:  Line 549 – what is the vehicle? Please clarify early on in the methods section.

Response to comment 6: We appreciate the reviewer’s suggestion. The vehicle used in this study was 0.5% carboxymethylcellulose (CMC) in distilled water, which was used to prepare uniform suspensions of both prednisolone and PZT for oral administration. This choice was based on the following considerations:

  1. It is orally safe and well-tolerated in rodents over repeated dosing periods.
  2. It allows for uniform suspension and stable dispersion of both prednisolone and PZT.
  3. It is chemically inert and does not interfere with inflammatory or oxidative stress markers, ensuring the integrity of biological outcome measurements.

This information has been clarified in the Materials and Methods section to enhance methodological transparency.

Results:

Comment 7:  Define abbreviations used in all figures and tables. For example, Figure 6 and 7 should have CAT, SOD and GSH-Px abbreviated

Response to comment 7: Thank you for your observation. We have revised all figure legends to ensure that abbreviations are clearly defined at their first appearance. Specifically, abbreviations such as CAT (catalase), SOD (superoxide dismutase), and GSH-Px (glutathione peroxidase), as well as others including SIRT1 (sirtuin 1), NF-κB (nuclear factor-kappa B), TNF-α (tumor necrosis factor-alpha), IL-6 (interleukin-6), TGF-β (transforming growth factor-beta), MDA (malondialdehyde), and PZT (phytosome-encapsulated Zea mays L. var. ceratina tassel extract) have been consistently defined in the figure captions. This clarification ensures clarity and consistency throughout the results section.

Comment 8:  Line 89-90 Why were the phenolic compounds selected were not mentioned in the introduction?

Response to comment 8: Thank you for your helpful suggestion. We have revised the Introduction section to include specific mention of the phenolic compounds found in Zea mays L. var. ceratina tassel. The updated text now highlights key flavonoids such as anthocyanins, rutin, quercetin, and kaempferol, which are known for their antioxidant and anti-inflammatory activities and are relevant to the proposed protective mechanisms. This addition provides a clearer rationale for selecting these compounds in the study.

Natural compounds derived from plant sources have shown potential in modulating epigenetic regulators, such as SIRT1. Zea mays L. var. ceratina (purple waxy corn) is particularly rich in phytochemicals, especially flavonoids, including anthocyanins [8], rutin, quercetin, and kaempferol, which possess strong antioxidant and anti-inflammatory properties. While the kernels of this variety have been extensively studied, the tassel—a traditionally underutilized by-product—also contains a diverse array of bioactive compounds that may contribute to cardiovascular protection [8,9].

Discussion:

Comment 9:  Line 229 -230 The first sentence of the summary should focus on the effects of PZT and not PM2.5

Response to comment 9: Thank you for pointing this out. In response, we have revised the summary sentence to begin with the effects of PZT rather than PM2.5. This adjustment places proper emphasis on the intervention and better aligns with the study’s objectives.

“In summary, treatment with PZT significantly attenuated PM2.5-induced inflammation by reducing the expression of key inflammatory markers, including NF-κB, TNF-α, and IL-6 in cardiac tissues, as well as NF-κB, TNF-α, and TGF-β in vascular tissues. The most pronounced anti-inflammatory effects were observed at higher PZT doses. Prednisolone also reduced these markers, supporting the anti-inflammatory potential of PZT. These findings highlight the efficacy of PZT in mitigating PM2.5-induced cardiovascular inflammation.”

Comment 10: How does this study compare to human trials if any?

Response to comment 10: Thank you for your insightful comment. Currently, there are limited human trials specifically examining the cardiovascular protective effects of the PZT. However, several human studies have investigated the benefits of related flavonoids such as quercetin, rutin, and anthocyanins—key constituents of this extract—in reducing inflammation and oxidative stress. Our study contributes to this growing body of evidence by providing mechanistic insights using a well-established animal model, particularly highlighting the epigenetic modulation via SIRT1 activation. These findings form a scientific foundation for future translational research and the potential development of clinical studies to evaluate the efficacy and safety of PZT in human populations exposed to air pollution. As suggested in the manuscript, future studies could further explore sirtuin-regulated gene networks, determine optimal dosing strategies, and assess long-term efficacy under chronic exposure conditions to better support real-world applicability.

References:

  1. Alharbi, H. O. A., Alshebremi, M., Babiker, A. Y., & Rahmani, A. H. (2025). The Role of Quercetin, a Flavonoid in the Management of Pathogenesis Through Regulation of Oxidative Stress, Inflammation, and Biological Activities. Biomolecules15(1), 151. https://doi.org/10.3390/biom15010151
  2. Bazyar, H., Zare Javid, A., Ahangarpour, A., Zaman, F., Hosseini, S. A., Zohoori, V., Aghamohammadi, V., Yazdanfar, S., & Ghasemi Deh Cheshmeh, M. (2023). The effects of rutin supplement on blood pressure markers, some serum antioxidant enzymes, and quality of life in patients with type 2 diabetes mellitus compared with placebo. Frontiers in nutrition10, 1214420. https://doi.org/10.3389/fnut.2023.1214420
  3. Laudani, S., Godos, J., Di Domenico, F. M., Barbagallo, I., Randazzo, C. L., Leggio, G. M., Galvano, F., & Grosso, G. (2023). Anthocyanin Effects on Vascular and Endothelial Health: Evidence from Clinical Trials and Role of Gut Microbiota Metabolites. Antioxidants (Basel, Switzerland)12(9), 1773. https://doi.org/10.3390/antiox12091773 

Thank you once again for your valuable feedback. We appreciate the time and effort invested by the reviewers and editor in evaluating our manuscript. We have carefully addressed each point raised and made necessary revisions accordingly. We eagerly await further feedback and guidance from the editorial team.

Yours sincerely,

Nut Palachai

Round 2

Reviewer 1 Report

Comments and Suggestions for Authors

I have reviewed the revised manuscript and the authors' elaborate responses to the previous comments. The authors have adequately responded to most of the issues and have made all necessary corrections and clarifications throughout the manuscript, but I am concerned that this manuscript plagiarism report indicates a similarity index of 35%. This level of text duplication is much higher than the generally accepted threshold for original research articles. I recommend that the authors thoroughly review the duplications identified, ensure that all such borrowed material is adequately cited or paraphrased, and resubmit a revised manuscript with a much lower similarity index. These issues need to be resolved in order to uphold the integrity and academic value of this journal.
Therefore, I propose this article for minor revision prior to publication in this renowned journal.

Author Response

Response to reviewer and editor suggestion

We sincerely thank you for your letter and the reviewers’ insightful comments on our manuscript, SIRT1-Mediated Epigenetic Protective Mechanisms of Phytosome-Encapsulated Zea mays L. var. ceratina Tassel Extract in a Rat Model of PM2.5-Induced Cardiovascular Inflammation (Manuscript ID: ijms-3668623).

We appreciate the opportunity to revise our manuscript and are grateful for the constructive feedback. In response, we have carefully addressed the comments and made the necessary revisions to improve the clarity and scientific rigor of our work. Below, we provide a detailed account of the main corrections and our responses to the reviewers’ suggestions.

Response to reviewer 1
Comment: I have reviewed the revised manuscript and the authors' elaborate responses to the previous comments. The authors have adequately responded to most of the issues and have made all necessary corrections and clarifications throughout the manuscript, but I am concerned that this manuscript plagiarism report indicates a similarity index of 35%. This level of text duplication is much higher than the generally accepted threshold for original research articles. I recommend that the authors thoroughly review the duplications identified, ensure that all such borrowed material is adequately cited or paraphrased, and resubmit a revised manuscript with a much lower similarity index. These issues need to be resolved in order to uphold the integrity and academic value of this journal.

Therefore, I propose this article for minor revision prior to publication in this renowned journal.

Response: We sincerely appreciate the reviewer’s thoughtful comment and the opportunity to address this important concern.

We acknowledge that the initial similarity index of the manuscript was reported as 34% using Turnitin. Upon detailed review of the highlighted content, we found that much of the similarity stemmed from necessary and standardized scientific terminology. This includes treatment doses (100, 200, and 400 mg/kg BW, which presented in our publication), the scientific name Zea mays L. var. ceratina, and commonly used biochemical and molecular terms such as nuclear factor-kappa B (NF-κB), tumor necrosis factor-alpha (TNF-α), interleukin-6 (IL-6), transforming growth factor-beta (TGF-β), sirtuin 1 (SIRT1), superoxide dismutase (SOD), catalase (CAT), glutathione peroxidase (GSH-Px), and malondialdehyde (MDA). Additionally, the names of chemical reagents, assay kits, and commercial products cannot be paraphrased without compromising clarity or scientific accuracy.

Nonetheless, we have thoroughly revised the entire manuscript—not just the Materials and Methods section—to minimize textual similarity while preserving the scientific integrity of the work. After these revisions, we re-evaluated the manuscript using Turnitin (excluding the reference list), and the similarity index was significantly reduced to 17%, which is well within the acceptable threshold for original research articles.

We also confirm that any remaining similarities—unavoidable due to technical, scientific, or chemical nomenclature—have been adequately cited where appropriate.

For your reference, we have attached the updated Turnitin similarity report after revision to demonstrate the improvements made.

We are confident that these revisions address the reviewer’s concern and further enhance the clarity, originality, and academic integrity of our manuscript. We sincerely thank the reviewer for the valuable feedback and hope the revised version meets the journal’s standards for publication.

Yours sincerely,

Nut Palachai
